# Effects of noise on integration of acoustic and electric hearing within and across ears

**Shelby Willis[1], Brian C. J. Moore[2], John J. Galvin III[3], Qian-Jie Fu[1]***

**1** Department of Head and Neck Surgery, David Geffen School of Medicine, University of California Los Angeles, Los Angeles, California, United States of America, **2** Department of Experimental Psychology, University of Cambridge, Cambridge, United Kingdom, **3** House Institute Foundation, Los Angeles, California, United States of America

* qfu@mednet.ucla.edu

## Abstract

In bimodal listening, cochlear implant (CI) users combine electric hearing (EH) in one ear and acoustic hearing (AH) in the other ear. In electric-acoustic stimulation (EAS), CI users combine EH and AH in the same ear. In quiet, integration of EH and AH has been shown to be better with EAS, but with greater sensitivity to tonotopic mismatch in EH. The goal of the present study was to evaluate how external noise might affect integration of AH and EH within or across ears. Recognition of monosyllabic words was measured for normal-hearing subjects listening to simulations of unimodal (AH or EH alone), EAS, and bimodal listening in quiet and in speech-shaped steady noise (10 dB, 0 dB signal-to-noise ratio). The input/output frequency range for AH was 0.1–0.6 kHz. EH was simulated using an 8-channel noise vocoder. The output frequency range was 1.2–8.0 kHz to simulate a shallow insertion depth. The input frequency range was either matched (1.2–8.0 kHz) or mismatched (0.6–8.0 kHz) to the output frequency range; the mismatched input range maximized the amount of speech information, while the matched input resulted in some speech information loss. In quiet, tonotopic mismatch differently affected EAS and bimodal performance. In noise, EAS and bimodal performance was similarly affected by tonotopic mismatch. The data suggest that tonotopic mismatch may differently affect integration of EH and AH in quiet and in noise.

## Introduction

Despite considerable efforts over the last 30 years, advances in cochlear implant (CI) technology and signal processing have yet to show substantial gains in speech performance. While high stimulation rates, deeply inserted electrodes, and current focusing all offer theoretical advantages over previous technology, none have shown consistent advantages for speech perception [1–3]. The poor functional spectral resolution and limited temporal information provided by CIs continues to limit the perception of speech in noise, speech prosody, vocal emotion, tonal language, and music [4–10]. Arguably, one of the greatest improvements in CI outcomes has come from combining electric hearing (EH) from the CI with acoustic hearing (AH) in the same ear (electric-acoustic stimulation, or EAS) or in opposite ears (bimodal listening). Note that these definitions do not include BiEAS (residual AH in both ears, CI in one ear) or BiBiEAS (residual AH in both ears, CI in both ears). Residual AH provides detailed

**Data Availability Statement:** All relevant data are within its supporting information files.

**Funding:** The research was supported by The National Institute on Deafness and Other Communication Disorders (NIDCD) R01-DC-

016883. The funders had no role in study design, data collection and analysis, decision to publish, or preparation of the manuscript.

**Competing interests:** The authors have declared that no competing interests exist.

low-frequency information that can greatly benefit CI users under challenging listening conditions, including perception of speech in noise and music perception [11–23]. Most of these previous studies of combined AH and EH (AEH) have assessed bimodal listening.

With relaxed candidacy criteria and the advance of electrode designs and hearing preservation surgery, increasing numbers of CI users have some residual AH in the implanted ear, allowing for EAS. Several CI manufacturers have developed hybrid speech processors that integrate hearing aid (HA) amplification and CI signal processing in the same ear. For EAS CI users, the current spread associated with electric stimulation may interfere with AH at the periphery. For bimodal users, there is no peripheral interaction between AH and EH, which may be advantageous. Similar to bimodal listening, Gantz et al. [24] found better speech performance with EAS than with the CI alone.

For AEH, clinical fitting of the CI is often performed without regard to the characteristics of AH, which may limit the integration of AH and EH due to two major factors–energetic/ modulation interference and information interference. For EAS patients, the broad current spread with EH may interact with the spread of excitation (SOE) produced by AH [25]. This may lead to interference both in terms of energetic masking and of modulation masking, whereby the amplitude modulation (AM) patterns associated with acoustic and electric stimulation interfere with one another [26, 27]. For EAS, this interference occurs more at the periphery; for bimodal listening, such interference would necessarily be more central in origin. For bimodal and EAS patients, the same temporal envelope information may also be delivered to different cochlear locations within or across ears ("tonotopic mismatch").

Sound quality differences between AH and EH may also affect integration of acoustic and electric hearing. To our knowledge, such sound quality differences have not been systematically studied in CI users, in quiet or noise. If the peripheral representations are sufficiently different, it is likely that sound quality may also be different across ears. While not the same as sound quality measurements across ears, inter-aural pitch matching experiments have shown difficulties matching the pitch of an acoustic signal in one ear [which contains temporal fine-structure (TFS) information that is important to pitch perception] to the pitch associated with stimulation of a single electrode [which does not contain TFS information and stimulates a broader cochlear region]. Work by Reiss and colleagues [28–30] has shown that inter-aural frequency mismatch can limit binaural fusion of acoustic and electric stimulation across ears or bilateral electric stimulation. Such inter-aural frequency mismatch can also limit perception of inter-aural timing differences (ITDs) that are important for sound source localization and image fusion [31–33].

Other studies have shown advantages for combining non-overlapping spectral information across ears. For example, Kulkarni et al. [34] showed a binaural advantage in spectral resolution for AH listeners when spectral bands were combined across ears. Similarly, Aronoff et al. [35] found that, for bilateral CI listeners, spectral resolution was significantly better when distributing 12 spectral channels across ears than within ears. Kong and Braida [36] found that the benefit of combining low-frequency AH with contralateral broadband EH was greater for vowels than for consonants, for both real and simulated CI patients. Luo and Fu [37], testing Chinese Mandarin-speaking NH participants listening to bimodal CI simulations, found that acoustic information < 500 Hz was more important for lexical tone perception (where voice pitch cues are important), and that acoustic information > 500 Hz was important for Mandarin phoneme recognition. The authors also found no difference for speech understanding in noise whether or not there was inter-aural spectral overlap between the simulated AH and EH. "Zipper" CI signal processing, where independent electrode information is provided to each ear, has shown binaural advantages in temporal processing and/or speech performance relative to monaural performance [38, 39].

Fu et al. [40] measured recognition of vowels in quiet for normal-hearing (NH) subjects listening to simulations of residual AH and EH. NH subjects and simulations were used to explicitly control the extent of stimulation within the cochlea and to directly compare the perception of EAS and bimodal stimulation. Such comparisons cannot easily be made using real EAS and bimodal CI patients, as the extent/quality of residual AH and the electrode-neural interface (the number and position of intra-cochlear electrodes relative to healthy neurons) are likely to vary markedly across ears and/or CI users. In the Fu et al. study [40], results showed that AH and EH were better combined for EAS than for bimodal listening. Relative to simulated EH, the benefits of simulated AEH were nearly twice as large for EAS as for bimodal listening. Also, acoustic-electric integration efficiency (IE, defined as the ratio between observed and predicted AEH performance; see later for details) was generally better for EAS than for bimodal listening. EAS IE was significantly affected by tonotopic mismatch, while bimodal IE was not. For CI-only and bimodal listening, there appeared to be a tradeoff between the amount of speech information and tonotopic mismatch. Compared to bimodal listening, EAS was less affected by small amounts of tonotopic mismatch, but more affected by larger mismatches.

In a subsequent study, Fu et al. [41] varied the carriers used in the CI simulations, including broad-band noise (limited only by the channel bandwidth), narrow-band noise, and sine waves. Reducing the bandwidth significantly improved simulated EH and bimodal performance, but not EAS performance. These results suggest that the inherent amplitude fluctuations in a broad-band noise carrier may negatively affect integration of AH and EH across ears. While reducing the bandwidth may improve both bimodal performance and integration efficiency for speech in quiet, it is possible that the deleterious effects of the inherent noise fluctuations on bimodal performance may be smaller or even absent for speech in noise. The amplitude fluctuations in external noise may mask the inherent noise fluctuations in broad-band noise carriers used to simulate EH. External noise may also reduce sound quality differences between AH and EH, thus improving the integration of AH and EH across ears; this might be true for both CI simulations and the real CI case, especially for bimodal listening. While previous studies showed that bimodal benefits are highly variable across individual CI users and different listening tasks and conditions, bimodal benefits are generally greater for speech in noise than for speech in quiet [21]. The bimodal benefit for speech in noise has often been attributed to a better ability to segregate the speech from the noise using residual AH. If external noise can mask the inherent noise fluctuations in simulated EH, it is possible that integration of AH and EH across ears may be better than previously observed in quiet.

The main goal of the present study was to evaluate integration of AH and simulated EH within and across ears in the presence of noise. Monosyllabic word recognition was measured in NH subjects using simulations of unimodal (AH or EH), EAS, and bimodal listening for speech in quiet or in steady speech-shaped noise. We hypothesized that in noise, AH and EH would be better integrated within (EAS) than across ears (bimodal), but with greater sensitivity to tonotopic mismatch than with bimodal listening, similar to our previous results in quiet [40, 41]. We also expected that, similar to the greater benefits observed in noise than in quiet in real bimodal and EAS listeners [21], we expected integration of AH and EH to be better in noise than in quiet in the present CI simulations.

## Materials and methods

This study was approved by the Institutional Review Board of the University of California, Los Angeles (UCLA). Prior to participation, written informed consent was obtained from all subjects, in accordance with a protocol approved by the Institutional Review Board at UCLA.

## Subjects

Ten NH subjects (5 males and 5 females) participated in this study. Participants were recruited from UCLA main campus. Recruitment postings were placed on the Digest News & Events website at the UCLA David Geffen School of Medicine (https://digest.dgsom.ucla.edu/). Inclusion criteria was that subjects have normal AH in both ears for all audiometric frequencies, that they were at least 18 years old, and were native speakers of American English. The mean age at testing was 25 years (range: 23–30 years). All subjects had thresholds <20 dB HL for audiometric frequencies 250, 500, 1000, 2000, 4000, and 8000 Hz in each ear. While the subject sample is small, the normal AH status in each ear presumably reflected uniform neural survival. As such, the various spectral degradation manipulations could be applied with the expectation that the effects would be similar across subjects.

## Test stimuli and procedures

Monosyllable CNC words spoken by one male talker were used. Stimuli were delivered via circumaural headphones (Sennheiser HDA-200) connected to separate channels of a mixer (Mackie 402 VLZ3), which was connected to an audio interface (Edirol UA-EX25). Before signal processing, all stimuli were normalized to have the same long-term root-mean-square (RMS) energy. Word recognition was measured in quiet and in noise at two signal-to-noise ratios (SNRs): 10 dB and 0 dB. Steady noise was filtered to have same spectrum averaged across all words. For each condition, a CNC list was randomly selected and words were randomly selected from within the list (without replacement) and presented to the subject, who was asked to repeat what they heard as accurately as possible. The experimenter calculated the percent of words correctly identified. All words in the list were scored, resulting in a total of 50 words for each subject and condition. No training or trial-by-trial feedback was provided. The test order of different listening conditions and SNRs was randomized within and counter-balanced across subjects.

## Simulations

Residual AH was simulated by bandpass filtering the speech signal between 0.1 and 0.6 kHz (20[th] order Butterworth filters; 240 dB/octave). This range was selected to represent the residual hearing available to some EAS and bimodal CI listeners, and to convey some information about the first formant frequency in speech. EH was simulated using 8-channel noise vocoders, similar to those described by Shannon et al. [42]. The input frequency range was divided into 8 bands (4[th] order Butterworth filters; 48 dB/octave), distributed according to Greenwood's frequency-to-place formula [43]. The temporal envelope was extracted from each analysis band by half-wave rectification and low-pass filtering (4[th] order Butterworth filter with 160-Hz cutoff frequency). The temporal envelope from each channel was used to modulate corresponding noise bands; the filter slope for the noise band carriers was same as that of the analysis filters. The modulated noise-bands were summed and the output was adjusted to have the same long-term RMS energy as the input. The output frequency range of the noise vocoders was fixed at 1.2–8.0 kHz. The lowest output frequency (1.2 kHz) corresponds to the cochlear location for a 20-mm insertion of an electrode array according to Greenwood [43] and is slightly higher than the median upper edge frequency of the residual AH (approximately 1.1 kHz) for hybrid CI patients reported by Karsten et al. [44]. The highest output frequency (8.0 kHz) is similar to the highest input frequency commonly used in commercial CI speech processors. Note that the output range of the CI simulations was not intended to simulate specific commercial CI devices, which vary in terms of array length, the number of electrodes and electrode spacing. Rather, the output frequency range was fixed, and the input frequency range was varied to

present different amounts of acoustic information while introducing different amounts of tonotopic mismatch. The upper limit of the simulated CI input frequency range was always 8.0 kHz. The lower limit was 0.6 or 1.2 kHz. When the limit was 0.6 kHz, all speech information was presented via the combination of AH and EH, but with a place mismatch of 4.3 mm at the apical end of the simulated electrode array according to Greenwood's [43] function. This condition is denoted as the "EH-mismatched" frequency allocation. When the lower limit of the input frequency range was 1.2 kHz, less acoustic information was presented, but there was no place mismatch between AH and EH and no overlap between the AH and EH input frequency ranges. This condition is denoted as the "EH-matched" frequency allocation. Performance with the AH and EH simulations alone (unimodal) was measured with one ear only. For the EAS simulation, AH and EH simulations were delivered to one earpiece of the headphones. For the bimodal simulation, AH and EH simulations were delivered to opposite earpieces of the headphones. The overall presentation level for the stimuli (combined speech and noise) was 60 dBA.

It should be noted that the present simulations (and indeed, all CI simulations) do not capture potential deficits in speech processing at higher levels of the auditory system that may be impaired by long-term hearing loss. For example, previous studies have shown that for prolonged auditory deprivation, cortical deficits and/or slower cortical development that may persist even after cochlear implantation [45, 46]. Such central deficits were not captured by the present simulations, which were more intended to simulate different peripheral limitations, but perceived with normal hearing and (presumably) normal higher-level auditory processing.

## Results

Fig 1 shows mean word recognition scores with the AH, EH, bimodal, and EAS listening conditions as a function of SNR; the left panels show data with the EH-mismatched allocation and

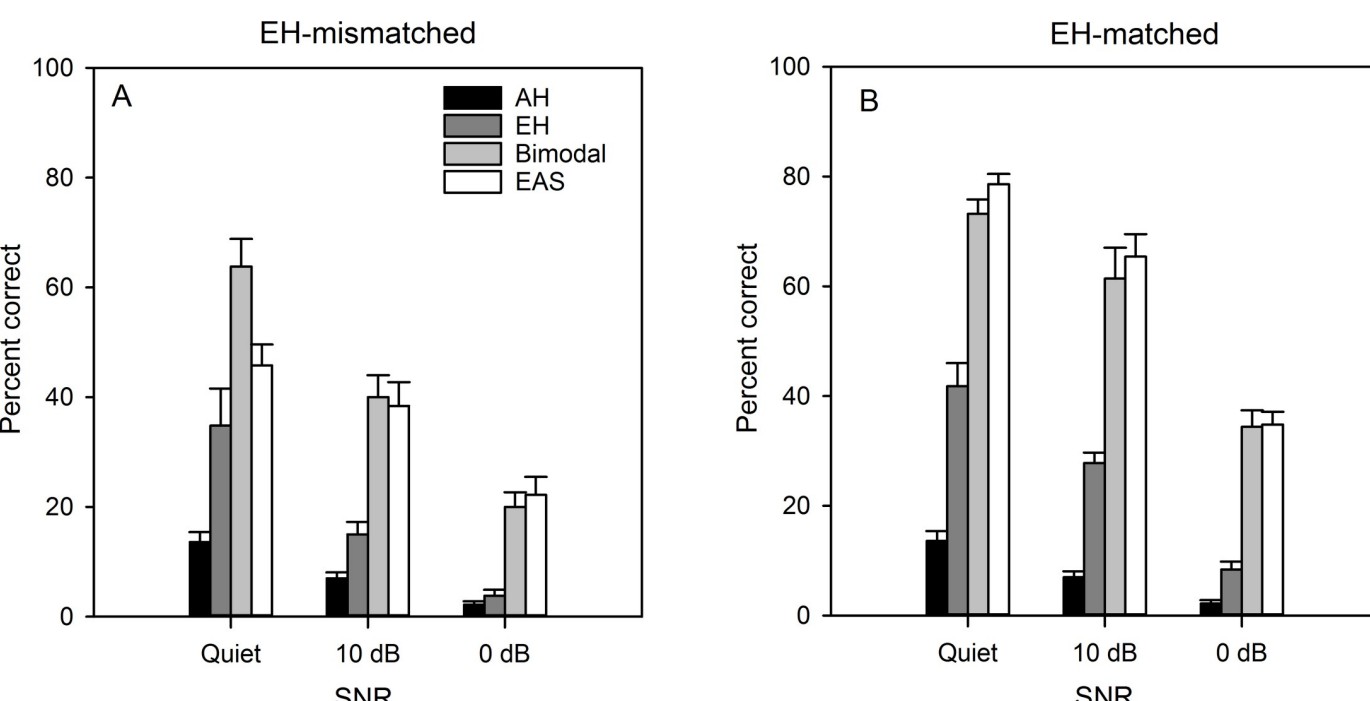

**Fig 1. Mean CNC word recognition for the EH-mismatched (Panel A) and the EH-matched frequency allocations (Panel B).** Data are shown for the AH, EH, bimodal, and EAS listening conditions. The error bars show the standard error.

the right panels show data with EH-matched allocation. Performance generally worsened as the SNR was reduced, and generally improved with AEH over AH or EH alone.

A repeated-measures analysis of variance (RM ANOVA) was performed on arc-sine transformed unimodal word recognition scores, with SNR (Quiet, 10 dB, 0 dB) and unimodal listening condition (AH, EH-mismatched, EH-matched) as factors; complete statistical results are shown in Table 1. Significant main effects were observed for SNR and unimodal listening condition ($P < 0.001$ in all cases); there was no significant interaction ($P = 0.071$). Another RM ANOVA was performed on the arcsine-transformed word recognition scores in Fig 1, this time with SNR (Quiet, 10 dB, 0 dB), CI mode (EH, bimodal, EAS), and CI allocation (mismatched, matched) as factors; complete statistical results are shown in Table 1. Significant main effects were observed for SNR, CI mode, and CI allocation ($P < 0.001$ in all cases). Significant interactions were observed between CI mode and CI allocation ($P = 0.011$), and among SNR, CI mode and CI allocation ($P = 0.033$).

The advantage for combined AH and EH ("AEH advantage") was calculated relative to EH-only performance. Fig 2 shows the AEH advantage for word recognition with bimodal or EAS listening (relative to EH performance) as a function of SNR. In quiet (Panel A), the AEH advantage for the EH-mismatched condition (black bars) was much larger for bimodal than for EAS listening. In contrast, the AEH advantage for the EH-matched condition (gray bars) was larger for EAS than for bimodal listening. In noise, the AEH advantage was generally larger for the EH-matched than for the EH-mismatched condition, with no strong differences between bimodal and EAS listening. AEH advantage was generally reduced as the SNR was reduced. A RM ANOVA was performed on the AEH advantage data shown in Fig 2, with SNR (Quiet, 10 dB, 0 dB), AEH mode (bimodal, EAS), and CI allocation (EH-mismatched, EH-matched) as factors. A significant main effect was observed only for CI allocation [dF(res) = 1(9); F = 36.3; $P < 0.001$; $\eta^2 = 0.80$]. However, there was a significant interaction among SNR, AEH mode, and CI allocation [dF(res) = 2(18); F = 5.1; $P = 0.017$; $\eta^2 = 0.36$], most likely driven by the strong difference in AEH according to CI allocation for EAS in the quiet listening condition.

Integration efficiency (IE) for AEH was calculated for the present bimodal and EAS data as the ratio between observed and predicted AEH performance, as in Fu et al. [40].

$$IE = P_{AEH}/\hat{P}_{AEH} \tag{1}$$

**Table 1. Statistical results for word recognition scores.**

| Unimodal | dF, res | F | P | $\eta^2$ | Observed power | Bonferroni post-hoc ($P < 0.05$) |
|---|---|---|---|---|---|---|
| SNR | 2,18 | 93.7 | *< 0.001 | 0.91 | > 0.99 | Quiet>10 dB>0 dB |
| Unimodal | 2,18 | 32.8 | *< 0.001 | 0.79 | > 0.99 | EH-matched>EH-mismatched>AH |
| SNR*Unimodal | 4,36 | 2.4 | 0.071 | 0.21 | 0.62 | |
| **CI mode** | **dF, res** | **F** | **P** | **$\eta^2$** | **Observed power** | **Bonferroni post-hoc ($P < 0.05$)** |
| SNR | 2,18 | 110.7 | *< 0.001 | 0.93 | > 0.99 | Quiet>10 dB>0 dB |
| CI mode | 2,18 | 175.8 | *< 0.001 | 0.95 | > 0.99 | Bimodal, EAS>EH |
| CI allocation | 1,9 | 51.9 | *< 0.001 | 0.85 | > 0.99 | Matched>Mismatched |
| SNR*CI mode | 4,36 | 1.0 | 0.399 | 0.10 | 0.30 | |
| SNR*CI allocation | 2,18 | 1.4 | 0.275 | 0.13 | 0.26 | |
| CI mode*CI allocation | 2,18 | 5.9 | *0.011 | 0.40 | 0.81 | Mismatched: Bimodal>EAS>EH; Matched: Bimodal, EAS>EH |
| SNR*CI mode*CI allocation | 4,36 | 3.0 | * 0.033 | 0.25 | 0.73 | Quiet, Mismatched: BI>EAS>EH; Quiet, Matched: BI, EAS>EH |

For the unimodal analysis in the top part of the table, factors included SNR (Quiet, 10 dB SNR, 0 dB SNR) and Unimodal listening (AH, EH-mismatched, EH-matched). For the CI mode analysis in the bottom part of the table, factors included SNR (Quiet, 10 dB SNR, 0 dB SNR), CI mode (EH, EAS, Bimodal), and CI allocation (mismatched, matched).

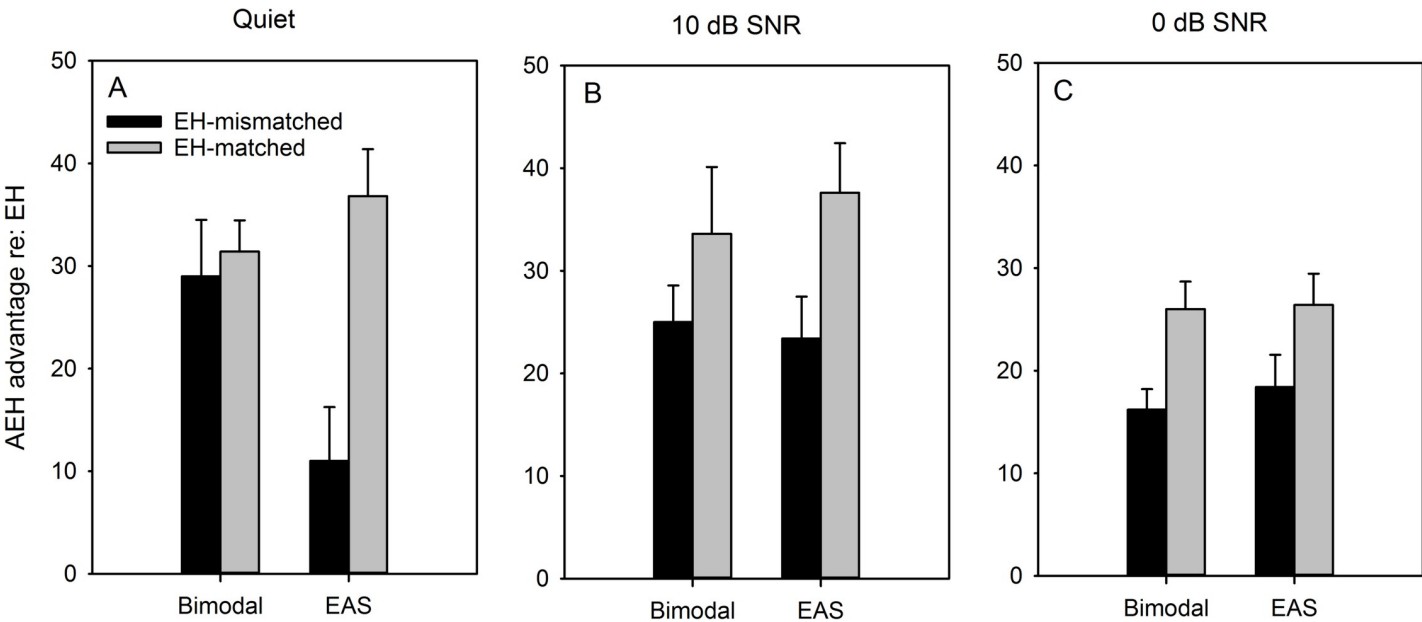

**Fig 2. Mean AEH advantage for bimodal and EAS listening (relative to EH performance) for CNC words in quiet (Panel A), at 10 dB SNR (Panel B), and at 0 dB SNR (Panel C).** Data are shown for the EH-mismatched (black bars) and EH-matched frequency allocations (gray bars). The error bars show the standard error.

where $P_{AEH}$ is the observed (measured) AEH proportion correct and $\hat{P}_{AEH}$ is the predicted AEH proportion correct. The predicted AEH proportion correct was based on the probability-summation rule:

$$\hat{P}_{AEH} = 1 - (1 - P_{AH})(1 - P_{EH}) \tag{2}$$

where $P_{AH}$ and $P_{EH}$ represent the observed (measured) proportions correct and $(1 - P_{AH})$ and $(1 - P_{EH})$ represent the observed proportions incorrect for AH alone and EH alone, respectively. According to this rule, the product $(1 - P_{AH})(1 - P_{EH})$ corresponds to the probability of being incorrect for both AH and EH. It should be noted that other methods for predicting the way that AH and EH are combined could give different outcomes [47]. However, the simple method defined by Eq 1 was considered sufficient for comparing IE across conditions. IE values $> 1$ imply "super-additive" or synergistic integration of acoustic and electric hearing.

Fig 3 shows IE as a function of SNR with bimodal (left panel) and EAS listening (right panel) with the EH-mismatched and EH-matched allocations. In all cases, IE increased as the SNR was reduced, largely due to the poorer AH and EH performance with increasing noise. In general, there was very little effect of listening condition or EH allocation on IE, except for the higher IE for the EH-matched allocation for EAS in the quiet listening condition. A RM ANOVA was performed on the IE data shown in Fig 3, with SNR (Quiet, 10 dB, 0 dB), AEH mode (binaural, EAS) and CI allocation (EH-mismatched, EH-matched) as factors. Note that data for 1 subject was excluded because of 0% correct scores for AH and EH at 0 dB SNR, which prohibited accurate estimation of IE. A significant main effect was observed only for SNR [dF(res) = 2(16); F = 19.1; $P < 0.001$; $\eta^2 = 0.70$], and there were no significant interactions. Post-hoc Bonferroni pairwise comparisons showed that IE was significantly larger at 0

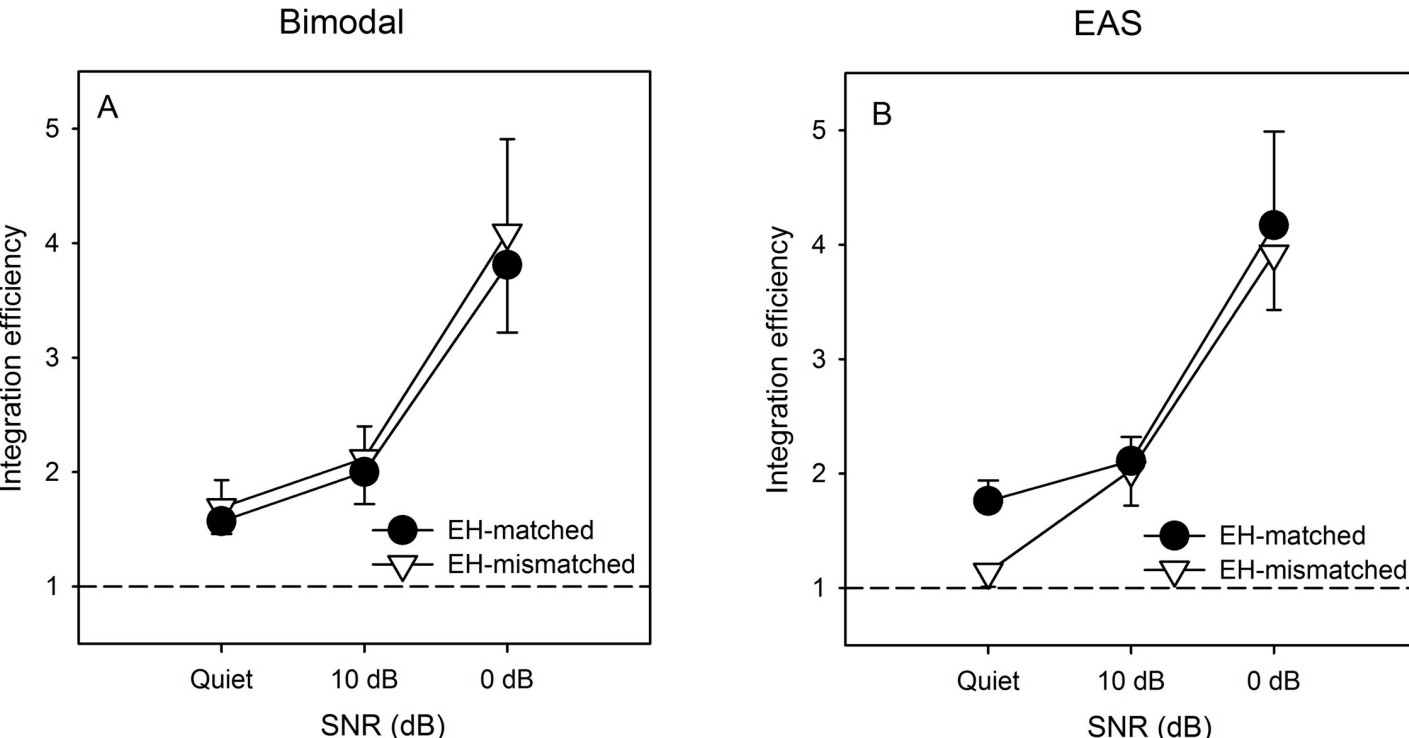

**Fig 3. Mean IE for CNC words as a function of SNR with the EH-matched (filled circles) and EH-mismatched frequency allocations (open circles) for bimodal (Panel A) and EAS listening (Panel B).** Values > 1 indicate that observed AEH performance was better than predicted by AH and EH performance. The error bars show the standard error.

dB SNR than in quiet or at 10 dB SNR ($P < 0.05$ in both cases), with no significant difference between quiet and 10 dB SNR.

## Discussion

Consistent with our previous studies [40, 41], CNC word recognition in quiet was poorer with EAS than with bimodal listening when there was a tonotopic mismatch in the simulated EH. Different from our previous studies, performance in quiet was similar with bimodal and EAS listening when there was no tonotopic mismatch. Contrary to our hypothesis, there was no difference between bimodal and EAS performance in noise, regardless of the degree of tonotopic mismatch in EH. The present data suggest that integration of AH and EH may be affected by interactions among the degree of tonotopic mismatch, the distribution of the AH (same or opposite ear of EH), and the presence of noise.

### Effect of outcome measures on integration of AH and EH

In Fu et al. [40, 41], integration of AH and EH was measured in quiet using a closed-set vowel recognition task, which was expected to be more sensitive to tonotopic mismatch than would be sentence recognition, where top-down processes might better accommodate the mismatch. For example, Fu et al. [48] showed stronger and more rapid adaptation to shifted frequency allocations in real CI users for recognition of sentences than words in quiet. In the present study, open-set CNC word recognition was measured at some "midway" point between closed-set vowel recognition (more bottom-up) and open-set sentence recognition (more top-down).

In quiet, the EAS advantage over bimodal listening observed for vowel recognition in Fu et al. [40, 41] was not observed for word recognition in the present study. Interestingly, EH performance (with or without mismatch) was comparable between word recognition in the present study and vowel recognition in Fu et al. [40, 41]. However, adding residual AH appeared to provide a greater benefit for word recognition than for vowel recognition. It is possible that adding the residual AH allowed for better perception of initial and final consonants in words than with EH alone. The vowel recognition task in Fu et al. [40, 41] required only perception of medial vowels, where low-frequency acoustic cues may have largely facilitated perception of some first formant cues. For real bimodal CI listeners, Yoon et al. [20] showed a greater AEH benefit for sentence recognition in quiet than for vowel or consonants, and a greater benefit for consonants than for vowels. Taken together, the present and previous data suggest that some outcome measures may be more sensitive than others to tonotopic mismatch in EH, as well as the benefits of adding residual AH, depending on the listening demands.

## Effect of tonotopic mismatch on integration of AH and EH

The present data suggest that the effects of tonotopic mismatch between AH and EH may be mitigated by the presence of noise. In quiet, word recognition was significantly better when the EH allocation was tonotopically matched (Fig 1; Table 1). However, there were some interactions between EH tonotopic mismatch and other test conditions. For example, a significant AEH advantage for EAS was observed only when the EH allocation was tonotopically matched (Fig 2). For bimodal listening, a significant AEH advantage was observed regardless of the degree of mismatch in the EH frequency allocation; with either allocation, the AEH benefit was significantly greater for bimodal than for EAS listening.

In noise, word recognition was significantly better when the EH allocation was tonotopically matched (Fig 1; Table 1). The effect of tonotopic mismatch on AEH advantage was similar for bimodal and EAS listening. The differential effects of mismatch on the distribution of AH (same or opposite ear as EH) observed in quiet were not observed in noise. As such, adding residual AH similarly benefitted bimodal and EAS listening in noise whether or not there was a tonotopic mismatch in EH. The present results in noise were different from those in quiet, and different from previous results in quiet from Fu et al. [40], suggesting that bimodal and EAS listening may be similarly sensitive to tonotopic mismatch in noise.

The present data show that the tradeoff between tonotopic matching and information loss observed in Fu et al. [40, 41] for vowel recognition in quiet persisted in noise. This is not consistent with Gifford et al. [49], who found that for real bimodal CI users, the best performance was observed when a wide input acoustic frequency range was used (more speech information, but with more mismatch), rather than a more tonotopically matched range (less information, but with less mismatch). For the present data, performance was always poorer with a tonotopic mismatch. Note that only one electrode insertion depth was used to create the EH tonotopic mismatch in the present study, compared to previous studies where a range of insertion depths/tonotopic mismatches were tested [40]. It is possible that the patterns of results may differ for words in noise (used in this study) and sentences in noise (used in Gifford et al. [49]), where top-down auditory processing might play a greater role. Also, real bimodal CI listeners have much greater experience with tonotopic mismatch in their device. Such adaptation to the mismatch in the EH simulation was not tested in the present study, as only acute performance was measured. It is possible that some short-term training may have reduced the effects of the mismatch [50–53]. It is also possible that the present CI simulations did not capture real CI users' performance.

### Effect of distribution of AH

While overall AEH was better with EAS than with bimodal listening in quiet, there was no significant difference between EAS and bimodal listening in noise, even when the SNR was relatively high (10 dB). This suggests that limited AH in either ear is beneficial in noise. It could be that the presence of AH in either ear improved perception of voicing information [16], rather than integration with the EH spectral pattern. The distribution of AH did not significantly affect AEH advantage with the EH-mismatched and EH-matched allocations. This suggests that AH did not necessarily facilitate spectral integration in noise in the same way as observed in quiet, where there was an advantage for EAS, but also less sensitivity to mismatch with bimodal listening.

### Effect of noise on integration of AH and EH

Not surprisingly, overall performance significantly worsened for all listening conditions as the amount of noise increased from 10 dB to 0 dB SNR (Fig 1; Table 1). The effects of tonotopic mismatch observed in quiet also persisted in noise, as performance was better with the EH-matched allocation despite the associated information loss. Significant interactions were observed between CI listening mode (EH, bimodal, EAS), and EH allocation only in quiet; here, bimodal performance was better than EAS when there was a tonotopic mismatch, but similar when there was no mismatch. In noise, there were no such interactions. This suggests that noise was a great equivocator on the differential effects of AH distribution (same or opposite ear from EH) and tonotopic mismatch observed in quiet. As such, bimodal and EAS may be similarly advantageous and similarly sensitive to tonotopic mismatch when listening to speech in noise.

It is possible that noise may have masked sound quality differences between residual AH and the noise-band vocoders used for the EH simulations, resulting in better fusion between AH and EH. However, noise caused performance to worsen for all unimodal (AH, EH) and AEH (bimodal, EAS) listening modes. As such, low-amplitude consonant and vowel information may have been masked by noise, which would obviate putative advantages due to consistent sound quality between AH and EH. This is somewhat borne out by the similar decrements in performance across listening modes as the SNR worsened. It is unclear whether a similar pattern of results would have been observed if sine-wave carriers (which have no inherent noise fluctuations) had been used instead of noise-band carriers.

### Effects of tonotopic mismatch, distribution of AH, and noise on IE

It is important to note that IE does necessarily not reflect absolute performance or even AEH performance gains per se. Instead, IE reflects the potential super-additivity of the unimodal components (AH, EH) for AEH listening under the various SNR, AH distribution, and tonotopic mismatch conditions. In general, the IE data show that as unimodal performance worsened due to increased noise and/or tonotopic mismatch, combining AH and EH contributed more strongly to performance gains. The present bimodal IE data are comparable to those from Yang and Zeng [54], who showed a mean value of 1.31 for concurrent vowel recognition for simulations of bimodal listening. IR was generally similar for the EH-matched and EH-mismatched allocations (except for in quiet, where there was a bimodal advantage for the EH-mismatched allocation). This reflects the greater sensitivity to tonotopic mismatch in quiet for EAS, and the greater sensitivity to mismatch at poor SNRs for bimodal listening.

The present IE data suggest that the relative contribution of AH to AEH listening increases as the listening conditions become more adverse (noise, tonotopic mismatch), regardless of how the AH is distributed across ears. Note that BiEAS listening were not simulated in this

study. It is possible that IE might have been even greater for BiEAS, as AH and EH might have only been marginally affected by bilateral presentation, but AEH performance would likely be even better than presently observed for bimodal or EAS listening. Gifford et al. [49] found a consistent advantage for BiEAS over bimodal listening in real CI users.

## Clinical implications and limitations

As indications for cochlear implantation continue to expand, and as surgical techniques and electrode designs continue to improve, combining AH and EH in the same ear, opposite ears, and even both ears will become more commonplace. Depending on the stability of residual AH and/or the electrode insertion depth, complete or partial residual AH preservation may be possible in the implanted ear. The present results suggest that minimizing tonotopic mismatch for EH may increase the benefit of AEH, especially for speech in noise, where both EAS and bimodal hearing were highly sensitive to tonotopic mismatch.

In the present study, the effects of tonotopic mismatch were acutely measured in NH participants listening to simulations of EH, bimodal and EAS, with no time for adaptation or training. Real CI patients can at least partly adapt to tonotopic mismatch with extended experience [48, 55–56]. CI simulation studies have shown that while listeners can automatically adapt to small amounts of tonotopic mismatch [57]; even greater adaptation is possible with explicit training [50–52]. It is possible that the effects of tonotopic mismatch observed in this CI simulation study might be mitigated over time in real CI users via passive adaptation and/or explicit auditory training.

To the extent that reducing tonotopic mismatch is advantageous for AEH, using radiological imaging to estimate the intra-cochlear electrode position, which then can be used to optimize frequency allocations [58, 59]. Intracochlear electrocochleography (ECochG), in which intra-cochlear electrodes are used to record responses to acoustic stimulation [60, 61], may also be used to estimate electrode position and guide tonotopic matching for CI patients with residual AH, especially for EAS listening.

One limitation of the current present study is that only a single frequency gap (0.6–1.2 kHz) between AH and EH was simulated; AH was limited to frequencies between 0.1 and 0.6 Hz, and the output frequency range for EH was 1.2–8.0 kHz. Increasing numbers of CI patients are being implanted with long electrode arrays and relatively deep insertions. Some CI patients also have residual AH in the implanted ear for frequencies above 0.6 kHz. For such CI patients, there may be no frequency gap or even an overlap between the frequency ranges covered by AH and EH. It is unclear how noise may affect integration of AH and EH in such cases. Also, there are increasing numbers of BiEAS CI users, as well as single-sided deaf CI users (normal AH in one ear, CI in the other ear). CI users' speech and localization performance has been shown to greatly benefit when there is limited AH in both ears [62, 63]. It is unclear how the availability of residual AH in both ears might have affected the present pattern of results with the present CI simulations. In future studies, it would be worthwhile to evaluate the effects of tonotopic mismatch in EH in the context of residual AH in both ears and CI in one or both ears.

Finally, it is unclear the extent to which the present CI simulation data extend to real bimodal and EAS CI users. CI simulations allow for better control of stimulation parameters, and NH listeners are likely to be a homogenous group in terms of performance, and have uniform auditory neural survival. This is rarely the case across CI users, who may use different devices, different parameter settings, different etiologies of deafness, different amounts of experience with their device, etc. Still, data from the present and previous CI simulation studies allow researchers to examine the effects of parameter manipulations in the "best-case

scenario" of normal AH. Aspects of CI patient variability (e.g., patchy neural survival, electrode insertion depth, the number of implanted electrodes) can also be simulated to better understand how CI signal processing may interact with patient-specific factors. The simulations also can be used to identify areas of interest that might be fruitful in optimizing stimulation parameters for real CI patients. Future studies with real bimodal and EAS CI users may further elucidate the importance of tonotopic matching for combined acoustic and electric hearing.

## Summary and conclusions

The present study examined monosyllabic word recognition in quiet and in noise for NH participants listening to simulations of residual AH, EH, bimodal (residual AH and EH in opposite ears), and EAS (residual AH and EH in the same ear). The EH simulations were either tonotopically matched or mismatched. Key findings include:

1. Overall performance was poorer when there was a tonotopic mismatch in EH In quiet, bimodal and EAS performance was similar when there was no tonotopic mismatch, but poorer for EAS than for bimodal when there was a mismatch. In noise, performance was similar between bimodal and EAS regardless of the degree of tonotopic mismatch.

2. In quiet, the AEH advantage relative to EH differed between bimodal and EAS, especially when there was a tonotopic mismatch, where performance was much poorer with EAS than with bimodal listening. In noise, the AEH advantage was similar for bimodal and EAS, regardless of the degree of tonotopic mismatch. However, the AEH advantage reduced with increasing noise level.

3. IE was generally similar between EAS and bimodal listening, and was largely unaffected by tonotopic mismatch.

## Supporting information

**S1 File. S1 File contains the raw word percent correct data for all subjects and listening conditions.**
(XLSX)

## Acknowledgments

We thank all participants for their time and effort. We also thank the editor and three anonymous reviewers for helpful comments.

## Author Contributions

**Conceptualization:** Qian-Jie Fu.

**Data curation:** Shelby Willis, Qian-Jie Fu.

**Formal analysis:** Shelby Willis, Brian C. J. Moore, John J. Galvin III, Qian-Jie Fu.

**Funding acquisition:** Qian-Jie Fu.

**Investigation:** Shelby Willis, Brian C. J. Moore, John J. Galvin III, Qian-Jie Fu.

**Methodology:** Brian C. J. Moore, Qian-Jie Fu.

**Project administration:** Qian-Jie Fu.

**Resources:** Qian-Jie Fu.

**Software:** Qian-Jie Fu.

**Supervision:** Qian-Jie Fu.

**Validation:** Qian-Jie Fu.

**Visualization:** Qian-Jie Fu.

**Writing – original draft:** Shelby Willis, Brian C. J. Moore, John J. Galvin III, Qian-Jie Fu.

**Writing – review & editing:** Shelby Willis, Brian C. J. Moore, John J. Galvin III, Qian-Jie Fu.

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
