## [Decision Letter · Decision Letter 0]

18 Aug 2020

PONE-D-20-21206

Effects of noise on the integration of acoustic and electric hearing within and across ears

PLOS ONE

Dear Dr. Fu,

Thank you for submitting your manuscript to PLOS ONE. After careful consideration, we feel that it has merit but does not fully meet PLOS ONE’s publication criteria as it currently stands. Therefore, we invite you to submit a revised version of the manuscript that addresses the points raised during the review process.

This manuscript is structured well with useful findings on the simulated electric hearing (EH) and acoustic hearing (AH) in the presence of noise in both the cases, hearing preservation implantation, and bimodal hearing. Please see the Editor's and reviewers' comments below this email.

We look forward to receiving your revised manuscript.

Kind regards,

Hussain Md Abu Nyeem, Ph.D.

Academic Editor

PLOS ONE

Additional Editor Comments:

While the presented findings on the effect of tonotopic mismatch hold promises for both the acoustic and electric hearing (AEH), its effect on the electric-acoustic stimulation (EAS) and bimodal hearing in the presence of noise appear to be slightly obvious. In other words, the presence of noise is more likely to contribute to the tonotopic mismatch, and thus, specific noises can be distinguished by their effect to make a more definite sense here. For example, we may want to know- how are the specific noise (with its properties or model) responsible for both EAS and bimodal hearing to be ‘highly’ sensitive to tonotopic mismatch? With the analysis of a few typical noise types in bimodal listening, the sensitivity variation of EAS and bimodal hearing thus may be contrasted to conclude their trend of sensitivity variation in noisy conditions.

Journal Requirements:

2. In your Methods section, please provide additional information about the demographic details of your participants. Please ensure you have provided sufficient details to replicate the analyses such as: a)  a description of any inclusion/exclusion criteria that were applied to participant inclusion in the analysis, b) a table of relevant demographic details, c) a statement as to whether your sample can be considered representative of a larger population, d) how participants were recruited to the study. In addition, please ensure you have described the statistical analyses within your Methods section.

"NO"

"NO"

Reviewers' comments:

Reviewer's Responses to Questions

**Comments to the Author**

1. Is the manuscript technically sound, and do the data support the conclusions?

Reviewer #1: Yes

Reviewer #2: Yes

Reviewer #3: Yes

2. Has the statistical analysis been performed appropriately and rigorously? 

Reviewer #1: Yes

Reviewer #2: Yes

Reviewer #3: Yes

3. Have the authors made all data underlying the findings in their manuscript fully available?

Reviewer #1: Yes

Reviewer #2: Yes

Reviewer #3: Yes

4. Is the manuscript presented in an intelligible fashion and written in standard English?

Reviewer #1: Yes

Reviewer #2: Yes

Reviewer #3: Yes

5. Review Comments to the Author

Reviewer #1: The simulation of CI signal processing tested on listeners with normal hearing serves as one method for evaluating speech understanding when the speech signal has been distorted using the noise vocoder. This assumes that the auditory system beyond the cochlea is "normal" and not affected by whatever oto-toxic agent led to the hearing loss. Thus, these simulations may, or may not, be good benchmarks for evaluation of such processing schemes. This caveat is rarely discussed in papers such as this one that are designed to evaluate speech processors.

Another small "flaw" is the lack of a reference for speech processing by listeners with normal hearing when the speech has been corrupted or distorted in other ways. For example, what level of performance would be expected if the noise vocoder were eliminated and only band pass filtering and additive noise were used to degrade the speech signals? Would similar results be obtained at SNRs below zero? Are there other models for combining information across channels that could be evaluated and used for comparison with these results?

This manuscript presents useful and timely results that were obtained from carefully designed and conducted experiments. A broader discussion of the implications of the work would be helpful.

Reviewer #2: This manuscript provides a detailed analysis of speech perception results for normal hearing subjects listening to simulations of 'electrical hearing' (EH) as applied in cochlear implant devices, and combinations of low frequency acoustic hearing combined with electrical hearing in the same ear (AEH) and opposite ear (bimodal). Monosyllabic word and phoneme scores are tabulated for quiet and for two competing noise conditions for ten young subjects. Two versions of the acoustic input were assessed in an attempt to elucidate the effect of 'tonotopic mismatch' between the electrical and acoustic hearing. The introduction provides appropriate background to the study and the decisions regarding the technical parameters are well justified. The procedures are clearly described and statistical analyses appear appropriate and comprehensive. I had some concerns about the applicability of simulation studies to the real world clinical environment but this is not overplayed by the authors and the limitations are clearly documented. There are a few typos along the way but the paper is generally well-written. Some parts of the discussion are a little difficult to follow and could be reworked somewhat for clarity. Most of the discussion provides reasonable conjectures about the main results that can be summarized as follows: 1. Low frequency acoustic hearing may provide improved speech perception when added to simulated cochlear implant hearing in either the ipsilateral (AEH) or contralateral ear (bimodal). 2. Tonotopic mismatch was found to have a more detrimental effect for speech perception in quiet for AEH than bimodal simulated listening. This effect appeared to decrease with increasing noise. The main conclusion that 'overall performance was poorer when there was a tonotopic mismatch' seems to be overstating the actual results which showed a loss of advantage for AEH (identified as EAS in the final summary - could be better to stick with the same terminology), but not for bimodal simulations or for AEH in noise. Despite my mainly positive comments relating to this manuscript, I have two issues that I believe need to be reworked. These both relate to the analysis of phoneme and word scores in this study and some conclusions that I believe are not justified. Firstly, by definition, word scores must be less than phoneme scores in this type of testing and the relationship is quite deterministic within the experimental error of these measures. Boothroyd and Nittrouer (1988 JASA 84:101-114) used a probability theory model of speech perception to look at contextual effects and showed that the relationship between phoneme and word scores in monosyllables is of the form - word score/100 = phoneme score/100^E, where E, the exponent, is approximately 3 for nonsense words and approximately 2.4 for real words. The exponent reflects the effective number of individual parts (phonemes) that need to identified to identify the word, and the difference between 3 for nonsense, and 2.4 for real words reflects the lexical context effects - ie., the increased probability of guessing a real word correctly compared to nonsense. A recent study has duplicated this result using real clinical data from cochlear implant users (Au, et al, 2018, Hearing Across the Lifespan (HEAL) conference, Lake Como Italy. see attachment). This relationship between phoneme and word scores in monosyllabic testing means that not only will phoneme scores always be lower than word scores (by differing amounts for different scores as the relationship is non-linear) but changes in word scores from one condition to another will predict changes in phonemes scores for the same conditions fairly accurately. The relevance to this study is that phoneme scores being significantly greater than word scores is claimed at numerous points as being an outcome, but it is not an outcome of the study, it is just a consequence of the more or less fixed relationship between such scores. In addition, another conclusion, that the integration efficiency (IE) factor was larger for phonemes than for words in some conditions does not really make sense as the scores (words and phonemes) are inextricably linked by the above relationship. I believe this outcome is a quirk of the mathematics and that, particularly for low scores, phoneme scores grow more rapidly than word scores (eg. phoneme scores have to improve from 0 to around 25% before word scores move from 0%). Note that the AEH advantages (fig.2) and the IE factors are derived from generally low word scores, particularly for SNR=0dB. I feel that the paper should be reworked removing the discussion and conclusions relating to different effects for word and phoneme scores.

Reviewer #3: This is a review of PONE-D-20-21206 entitled “Effects of noise on the integration of acoustic and electric hearing within and across ears.” The purpose of this study was to describe the integration of simulated electric hearing (EH) and acoustic hearing (AH) in the presence of noise within an ear, as is the case with hearing preservation implantation, or across ears (bimodal hearing). They investigated cases of simulated EH+AH with and without tonotopic mismatch. This manuscript was relatively straightforward, well written, and the results hold high clinical application. There are a few items requiring attention which would significantly improve the manuscript and its overall impact.

General comments:

A major concern with the study design and interpretation of results is that EAS conditions (EH+AH in the same ear) do not include integration of EH+AH across ears. However, most EAS patients also have acoustic hearing in the contralateral ear. As such, this would result in potential interference both peripherally (within an ear) and centrally (across ears). Though this isn’t a critical flaw, it is something that must be addressed as it impacts the clinical impact of the study. It would also be interesting to mention this as a point for additional study in future simulation studies.

Point-by-point comments:

Lines 75-77: An explanation here would be valuable regarding how qualitative differences impact integration (include references). Also, there is no mention here regarding binaural integration (particularly for binaural cues) and its influence on source segregation. As mentioned in “General comments,” both would be present with EAS listeners who were combining AH+EH within an ear and across ears simultaneously.

Line 97: “inherent” is included twice in this sentence

Lines 111-112: Could this also be pertinent for EAS (CI + binaural acoustic hearing)?

Figure 1: Red and green bars next to each other are difficult to distinguish for individuals with color blindness.

Lines 290-291: This should not be unexpected.

Lines 337-346: Real bimodal listeners typically have chronic listening experience in a “mismatch” condition. Also, a singular simulated insertion depth was used in the current study whereas most studies with real bimodal (or EAS) listeners include various different insertion depths. Both issues should be mentioned here as potential reasons for the discrepancy in findings.

6. PLOS authors have the option to publish the peer review history of their article (what does this mean?). If published, this will include your full peer review and any attached files.

Reviewer #1: No

Reviewer #2: No

Reviewer #3: No

<gdiv></gdiv>

---

## [Author Response · Author response to Decision Letter 0]

9 Sep 2020

To the editor:

Please find enclosed our revised submission “Effects of noise on integration of acoustic and electric hearing within and across ears”. 

We thank the editor and the reviewers and have greatly revised the MS with their comments in mind. Major changes include:

1. Removal of phoneme data. We had originally included the phoneme data because it offered better comparison to the vowel recognition data in our previous related studies. However, there seemed to be some confusion regarding percent correct, AEH advantage (the difference in performance when AH was added to EH, relative to EH-only), and IE [the ratio between observed AEH and predicted AEH (according to the AH and EH only performance]. It was always true that phoneme recognition was higher than word recognition. However, it seemed confusing that AEH advantage and IE might be greater for words than for phonemes. This had nothing to do with the relationship between phoneme and word recognition, but rather how adding AH to EH benefitted performance. As Rev. 2 points out, the lower overall AH and EH word scores allowed more “room for improvement.” But it appears that the phoneme data distracted from the main story, which was about how noise, the degree of tonotopic mismatch, and the distribution of AH across ears might affect integration of AH and EH. Accordingly, we have deleted all phoneme data, have revised the three figures, and have re-analyzed the data after excluding the phoneme data. We feel that this does not change the story at all, but may help reduce confusion.

2. We have expanded the Introduction and Discussion sections according to reviewers’ suggestions.

3. All figures are now in gray scale

We hope you find this revision acceptable, and let me know if you need further information.

Sincerely,

Qian-jie Fu

Additional Editor Comments:

While the presented findings on the effect of tonotopic mismatch hold promises for both the acoustic and electric hearing (AEH), its effect on the electric-acoustic stimulation (EAS) and bimodal hearing in the presence of noise appear to be slightly obvious. In other words, the presence of noise is more likely to contribute to the tonotopic mismatch, and thus, specific noises can be distinguished by their effect to make a more definite sense here. For example, we may want to know- how are the specific noise (with its properties or model) responsible for both EAS and bimodal hearing to be ‘highly’ sensitive to tonotopic mismatch? With the analysis of a few typical noise types in bimodal listening, the sensitivity variation of EAS and bimodal hearing thus may be contrasted to conclude their trend of sensitivity variation in noisy conditions.

>>Actually, we expected noise to interact with tonotopic mismatch with bimodal or EAS hearing similarly as in quiet (perhaps an “obvious” expectation). But this was not the case. There was a very strong effect of tonotopic mismatch for EAS in quiet, but not for bimodal listening. When noise was added, the effects of tonotopic mismatch were similar for EAS and bimodal listening. Noise does not contribute to tonotopic mismatch per se, but it appears to similarly mask the effects of tonotopic mismatch with EAS and bimodal listening. Of course, overall performance also declines with noise, whether or not there is a tonotopic mismatch. In the present study, speech and noise were combined before subsequent processing; as such the effects of noise within AH, EH-mismatched, and EH matched were spectrally limited within the bandwidth of each of these conditions. Different types of noise (multi-talker babble, gated noise, etc.) would be similarly processed by these frequency input/output function. It is possible that AEH benefit might have differed between steady noise (as used in the present study), competing speech or multi-talker babble. Perhaps a good comparison for future work….

 >>We have proofread the paper to ensure that the style conforms to PLOS One guidelines.

2. In your Methods section, please provide additional information about the demographic details of your participants. Please ensure you have provided sufficient details to replicate the analyses such as: a) a description of any inclusion/exclusion criteria that were applied to participant inclusion in the analysis, b) a table of relevant demographic details, c) a statement as to whether your sample can be considered representative of a larger population, d) how participants were recruited to the study. In addition, please ensure you have described the statistical analyses within your Methods section.

>> We have added: Ten NH subjects (5 males and 5 females) participated in this study. Participants were recruited from UCLA main campus. Recruitment postings were placed on the Digest News & Events website at the UCLA David Geffen School of Medicine (https://digest.dgsom.ucla.edu/). Inclusion criteria was that subjects have normal AH in both ears for all audiometric frequencies, that they were at least 18 years old, and were native speakers of American English. The mean age at testing was 25 years (range: 23-30 years). All subjects had thresholds <20 dB HL for audiometric frequencies 250, 500, 1000, 2000, 4000, and 8000 Hz in each ear. While the subject sample is small, the normal AH status in each ear presumably reflected uniform neural survival. As such, the various spectral degradation manipulations could be applied with the expectation that the effects would be similar across subjects.”

"NO"

a. Please clarify the sources of funding (financial or material support) for your study. List the grants or organizations that supported your study, including funding received from your institution.

d. If you did not receive any funding for this study, please state: “The authors received no specific funding for this work.”

>> We have added this information in the revised cover letter. 

"NO"

>> We have included this information in the revised cover letter. 

>>We have added the Supporting information file at the end of the MS.

5. Review Comments to the Author

Reviewer #1: The simulation of CI signal processing tested on listeners with normal hearing serves as one method for evaluating speech understanding when the speech signal has been distorted using the noise vocoder. This assumes that the auditory system beyond the cochlea is "normal" and not affected by whatever oto-toxic agent led to the hearing loss. Thus, these simulations may, or may not, be good benchmarks for evaluation of such processing schemes. This caveat is rarely discussed in papers such as this one that are designed to evaluate speech processors.

>>You make an important point…We have added to the end of the Simulations section of the Methods: It should be noted that the present simulations (and indeed, all CI simulations) do not capture potential deficits in speech processing at higher levels of the auditory system that may be impaired by long-term hearing loss. For example, previous studies have shown that for prolonged auditory deprivation, cortical deficits and/or slower cortical development that may persist even after cochlear implantation [45-46]. Such central deficits were not captured by the present simulations, which were more intended to simulate different peripheral limitations, but perceived with normal hearing and (presumably) normal higher-level auditory processing.”

Another small "flaw" is the lack of a reference for speech processing by listeners with normal hearing when the speech has been corrupted or distorted in other ways. For example, what level of performance would be expected if the noise vocoder were eliminated and only band pass filtering and additive noise were used to degrade the speech signals? Would similar results be obtained at SNRs below zero? Are there other models for combining information across channels that could be evaluated and used for comparison with these results?

>>We have added: “Other studies have shown advantages for combining non-overlapping spectral information across ears. For example, Kulkarni et al. [34] showed a binaural advantage in spectral resolution for AH listeners when spectral bands were combined across ears. Similarly, Aronoff et al. [35] found that, for bilateral CI listeners, spectral resolution was significantly better when distributing 12 spectral channels across ears than within ears. Kong and Braida [36] found that the benefit of combining low-frequency AH with contralateral broadband EH was greater for vowels than for consonants, for both real and simulated CI patients. Luo and Fu [37], testing Chinese Mandarin-speaking NH participants listening to bimodal CI simulations, found that acoustic information < 500 Hz was more important for lexical tone perception (where voice pitch cues are important), and that acoustic information > 500 Hz was important for Mandarin phoneme recognition. The authors also found no difference for speech understanding in noise whether or not there was inter-aural spectral overlap between the simulated AH and EH. “Zipper” CI signal processing, where independent electrode information is provided to each ear, has shown binaural advantages in temporal processing and/or speech performance relative to monaural performance [38-39].”

This manuscript presents useful and timely results that were obtained from carefully designed and conducted experiments. A broader discussion of the implications of the work would be helpful.

>>We have somewhat expanded the “Clinical implications and limitations” section of the Discussion

Reviewer #2:

There are a few typos along the way but the paper is generally well-written. 

>>We have carefully proof-read the revision and hopefully have removed typos and grammatical errors.

Some parts of the discussion are a little difficult to follow and could be reworked somewhat for clarity. 

>>Given that we have removed Discussion of the phoneme data (see below), we have greatly revised the Results and Discussion.

Despite my mainly positive comments relating to this manuscript, I have two issues that I believe need to be reworked. These both relate to the analysis of phoneme and word scores in this study and some conclusions that I believe are not justified. Firstly, by definition, word scores must be less than phoneme scores in this type of testing and the relationship is quite deterministic within the experimental error of these measures. Boothroyd and Nittrouer (1988 JASA 84:101-114) used a probability theory model of speech perception to look at contextual effects and showed that the relationship between phoneme and word scores in monosyllables is of the form - word score/100 = phoneme score/100^E, where E, the exponent, is approximately 3 for nonsense words and approximately 2.4 for real words. The exponent reflects the effective number of individual parts (phonemes) that need to identified to identify the word, and the difference between 3 for nonsense, and 2.4 for real words reflects the lexical context effects - ie., the increased probability of guessing a real word correctly compared to nonsense. A recent study has duplicated this result using real clinical data from cochlear implant users (Au, et al, 2018, Hearing Across the Lifespan (HEAL) conference, Lake Como Italy. see attachment). This relationship between phoneme and word scores in monosyllabic testing means that not only will phoneme scores always be lower than word scores (by differing amounts for different scores as the relationship is non-linear) but changes in word scores from one condition to another will predict changes in phonemes scores for the same conditions fairly accurately. The relevance to this study is that phoneme scores being significantly greater than word scores is claimed at numerous points as being an outcome, but it is not an outcome of the study, it is just a consequence of the more or less fixed relationship between such scores. 

>>” This relationship between phoneme and word scores in monosyllabic testing means that not only will phoneme scores always be lower than word scores…” Maybe a typo, but word scores should always be lower than phoneme scores, as you point out above, as is shown in your poster, and is shown in our data. We believe there may be some confusion regarding the relationships among word recognition, phoneme recognition, AEH advantage (the difference in performance when AH was added to EH), and IE (the ratio between the observed and predicted performance). AEH and IE are both derivative measures that look at different aspects of how AH and EH are combined within or across ears. It is entirely possible that adding AH may have benefitted word recognition (which had lower EH-only scores) more than phoneme recognition (which had higher EH-only scores). IE was also much better for words than phonemes, due to the poorer AH and EH performance. Thus, while word recognition was always poorer than phoneme recognition, word recognition may have benefitted more greatly from the addition of AH due to the generally poorer performance. 

 We realize that there may be unintended confusion when comparing recognition scores, AEH, and IE between words and phonemes at the various SNR. We have decided to remove all mention of phoneme scores in the revision. We had originally included the phoneme scores as they were more comparable to the vowel recognition measures in our previous study in quiet. As the phoneme scores do not add much to the general pattern of results, we now only present word recognition scores, as well as AEH and IE for word recognition.

In addition, another conclusion, that the integration efficiency (IE) factor was larger for phonemes than for words in some conditions does not really make sense as the scores (words and phonemes) are inextricably linked by the above relationship. I believe this outcome is a quirk of the mathematics and that, particularly for low scores, phoneme scores grow more rapidly than word scores (eg. phoneme scores have to improve from 0 to around 25% before word scores move from 0%). Note that the AEH advantages (fig.2) and the IE factors are derived from generally low word scores, particularly for SNR=0dB. I feel that the paper should be reworked removing the discussion and conclusions relating to different effects for word and phoneme scores.

>>There must be some misunderstanding, because we clearly state on line 390 of the original MS: “As shown in Fig 3, IE was much stronger for words than for phonemes.” But as noted above, we have removed all mention of phoneme scores in the revision.

Reviewer #3: 

A major concern with the study design and interpretation of results is that EAS conditions (EH+AH in the same ear) do not include integration of EH+AH across ears. However, most EAS patients also have acoustic hearing in the contralateral ear. As such, this would result in potential interference both peripherally (within an ear) and centrally (across ears). Though this isn’t a critical flaw, it is something that must be addressed as it impacts the clinical impact of the study. It would also be interesting to mention this as a point for additional study in future simulation studies.

>>We have added to the “Clinical implications and limitations” section of the Discussion: Also, there are increasing numbers of BiEAS CI users, as well as single-sided deaf CI users (normal AH in one ear, CI in the other ear). CI users’ speech and localization performance has been shown to greatly benefit when there is limited AH in both ears [62-63]. It is unclear how the availability of residual AH in both ears might have affected the present pattern of results with the present CI simulations. In future studies, it would be worthwhile to evaluate the effects of tonotopic mismatch in EH in the context of residual AH in both ears and CI in one or both ears.”

Lines 75-77: An explanation here would be valuable regarding how qualitative differences impact integration (include references). Also, there is no mention here regarding binaural integration (particularly for binaural cues) and its influence on source segregation. As mentioned in “General comments,” both would be present with EAS listeners who were combining AH+EH within an ear and across ears simultaneously.

>>Note that “BiEAS” was not simulated in this study. To clarify, we have revised a section of the first paragraph as: “Arguably, one of the greatest improvements in CI outcomes has come from combining electric hearing (EH) from the CI with acoustic hearing (AH) in the same ear (electric-acoustic stimulation, or EAS) or in opposite ears (bimodal listening). Note that these definitions do not include BiEAS (residual AH in both ears, CI in one or both ears)].”

In response to the other comments, we have added: “Sound quality differences between AH and EH may also affect integration of acoustic and electric hearing. To our knowledge, such sound quality differences have not been systematically studied in CI users, in quiet or noise. If the peripheral representations are sufficiently different, it is likely that sound quality may also be different across ears. While not the same as sound quality measurements across ears, inter-aural pitch matching experiments have shown difficulties matching the pitch of an acoustic signal in one ear [which contains temporal fine-structure (TFS) information that is important to pitch perception] to the pitch associated with stimulation of a single electrode [which does not contain TFS information and stimulates a broader cochlear region]. Work by Reiss and colleagues [28-30] has shown that inter-aural frequency mismatch can limit binaural fusion of acoustic and electric stimulation across ears or bilateral electric stimulation. Such inter-aural frequency mismatch can also limit perception of inter-aural timing differences (ITDs) that are important for sound source localization and image fusion [31-33].”

Line 97: “inherent” is included twice in this sentence

>>Corrected

Lines 111-112: Could this also be pertinent for EAS (CI + binaural acoustic hearing)?

>>In this paper and many others, EAS refers to combined electric and acoustic hearing in the same ear. We have added to the Discussion: “Note that BiEAS listening were not simulated in this study. It is possible that IE might have been even greater for BiEAS, as AH and EH might have only been marginally affected by bilateral presentation, but AEH performance would likely be even better than presently observed for bimodal or EAS listening. Gifford et al. [49] found a consistent advantage for BiEAS over bimodal listening in real CI users.” And later: “Also, there are increasing numbers of BiEAS CI users, as well as single-sided deaf CI users (normal AH in one ear, CI in the other ear). CI users’ speech and localization performance has been shown to greatly benefit when there is limited AH in both ears [62-63]. It is unclear how the availability of residual AH in both ears might have affected the present pattern of results with the present CI simulations.”

Figure 1: Red and green bars next to each other are difficult to distinguish for individuals with color blindness.

>>Figure 1 now in gray scale

Lines 290-291: This should not be unexpected.

>>Revised “As expected, performance was significantly better for phoneme than for word recognition. However, in terms of AEH advantage, there was a significant interaction between outcome measure and the degree of tonotopic mismatch in EH.”

Lines 337-346: Real bimodal listeners typically have chronic listening experience in a “mismatch” condition. Also, a singular simulated insertion depth was used in the current study whereas most studies with real bimodal (or EAS) listeners include various different insertion depths. Both issues should be mentioned here as potential reasons for the discrepancy in findings.

>>We had somewhat discussed these issues in the “Clinical implications and limitations” section of the Discussion of the original MS, but we have also added here in response to the reviewer’s comments. Paragraph revised as: The present data show that the tradeoff between tonotopic matching and information loss observed in Fu et al. [40-41] for vowel recognition in quiet persisted in noise. This is not consistent with Gifford et al. [49], who found that for real bimodal CI users, the best performance was observed when a wide input acoustic frequency range was used (more speech information, but with more mismatch), rather than a more tonotopically matched range (less information, but with less mismatch). For the present data, performance was always poorer with a tonotopic mismatch. Note that only one electrode insertion depth was used to create the EH tonotopic mismatch in the present study, compared to previous studies where a range of insertion depths/tonotopic mismatches were tested [40]. It is possible that the patterns of results may differ for words in noise (used in this study) and sentences in noise (used in Gifford et al. [49]), where top-down auditory processing might play a greater role. Also, real bimodal CI listeners have much greater experience with tonotopic mismatch in their device. Such adaptation to the mismatch in the EH simulation was not tested in the present study, as only acute performance was measured. It is possible that some short-term training may have reduced the effects of the mismatch [50-53]. It is also possible that the present CI simulations did not capture real CI users’ performance.”

---

## [Decision Letter · Decision Letter 1]

2 Oct 2020

Effects of noise on integration of acoustic and electric hearing within and across ears

PONE-D-20-21206R1

Dear Dr. Fu,

We’re pleased to inform you that your manuscript has been judged scientifically suitable for publication and will be formally accepted for publication once it meets all outstanding technical requirements.

Kind regards,

Hussain Md Abu Nyeem, Ph.D.

Academic Editor

PLOS ONE

**Comments to the Author**

1. If the authors have adequately addressed your comments raised in a previous round of review and you feel that this manuscript is now acceptable for publication, you may indicate that here to bypass the “Comments to the Author” section, enter your conflict of interest statement in the “Confidential to Editor” section, and submit your "Accept" recommendation.

Reviewer #1: All comments have been addressed

Reviewer #2: All comments have been addressed

2. Is the manuscript technically sound, and do the data support the conclusions?

Reviewer #1: Yes

Reviewer #2: Yes

3. Has the statistical analysis been performed appropriately and rigorously? 

Reviewer #1: Yes

Reviewer #2: Yes

4. Have the authors made all data underlying the findings in their manuscript fully available?

Reviewer #1: Yes

Reviewer #2: Yes

5. Is the manuscript presented in an intelligible fashion and written in standard English?

Reviewer #1: Yes

Reviewer #2: Yes

6. Review Comments to the Author

Reviewer #1: (No Response)

Reviewer #2: My apologies to the authors for typographical errors in my review. Of course, word scores are always lower than phoneme scores - I should have checked my review more closely! Removing all mention of the phoneme scores is an interesting way of dealing with some of the confusing findings. Maybe I would have gone the other way and removed the word scores, but I will not argue about the approach. Thank you for your careful revision.

7. PLOS authors have the option to publish the peer review history of their article (what does this mean?). If published, this will include your full peer review and any attached files.

Reviewer #1: No

Reviewer #2: No

---

## [Editor Report · Acceptance letter]

6 Oct 2020

PONE-D-20-21206R1 

Effects of noise on integration of acoustic and electric hearing within and across ears 

Dear Dr. Fu:

I'm pleased to inform you that your manuscript has been deemed suitable for publication in PLOS ONE. Congratulations! Your manuscript is now with our production department. 

Kind regards, 

on behalf of

Dr. Hussain Md Abu Nyeem 

Academic Editor

PLOS ONE